# Deficiency of *PRKD2* triggers hyperinsulinemia and metabolic disorders

Yao Xiao [1,2], Can Wang[1,2], Jia-Yu Chen[1,2], Fujian Lu[1,2], Jue Wang[1,2], Ning Hou[1,2], Xiaomin Hu[1,2], Fanxin Zeng[1,2], Dongwei Ma[1,2], Xueting Sun[1,2], Yi Ding[1,2], Yan Zhang[1,2,3], Wen Zheng[1,2], Yuli Liu[1,2], Haibao Shang[1,2], Wenzhen Zhu[1,2], Chensheng Han[1,2], Yulin Zhang[1,2], Kunfu Ouyang[4], Liangyi Chen [1,2], Ju Chen[5], Rui-Ping Xiao[1,2,3], Chuan-Yun Li [1,2] & Xiuqin Zhang[1,2]

Hyperinsulinemia is the earliest symptom of insulin resistance (IR), but a causal relationship between the two remains to be established. Here we show that a protein kinase D2 (*PRKD2*) nonsense mutation (K410X) in two rhesus monkeys with extreme hyperinsulinemia along with IR and metabolic defects by using extreme phenotype sampling and deep sequencing analyses. This mutation reduces *PRKD2* at both the mRNA and the protein levels. Taking advantage of a *PRKD2-KO* mouse model, we demonstrate that *PRKD2* deletion triggers hyperinsulinemia which precedes to IR and metabolic disorders in the *PRKD2* ablation mice. *PRKD2* deficiency promotes β-cell insulin secretion by increasing the expression and activity of L-type $Ca^{2+}$ channels and subsequently augmenting high glucose- and membrane depolarization-induced $Ca^{2+}$ influx. Altogether, these results indicate that down-regulation of *PRKD2* is involved in the pathogenesis of hyperinsulinemia which, in turn, results in IR and metabolic disorders.

[1] Institute of Molecular Medicine, Peking University, Beijing 100871, China. [2] Beijing Key Laboratory of Cardiometabolic Molecular Medicine, Peking University, Beijing 100871, China. [3] State Key Laboratory of Biomembrane and Membrane Biotechnology, Peking-Tsinghua Center for Life Sciences, Beijing 100871, China. [4] Drug Discovery Center, Key Laboratory of Chemical Genomics, Peking University Shenzhen Graduate School, Shenzhen 518055, China. [5] Department of Medicine, University of California San Diego School of Medicine, La Jolla, CA 92093, USA. These authors contributed equally: Yao Xiao, Can Wang. Correspondence and requests for materials should be addressed to C.-Y.L. (email: chuanyunli@pku.edu.cn) or to X.Z. (email: zhangxq@pku.edu.cn)

The prevalence of metabolic diseases and their complications are reaching epidemic proportions and have become the major cause of cardiovascular morbidity and mortality worldwide[1,2]. Hyperinsulinemia has been implicated in the pathogenesis of metabolic diseases, such as metabolic syndrome (MetS), type 2 diabetes (T2D), and a variety of cardiovascular diseases[3–5]. A combination of metabolic and cardiovascular diseases (i.e., cardiometabolic disease) is the leading cause of death in the west and developing countries (http://www.who.int/en/). Identification and characterization of pathogenic factors and signaling pathways are needed urgently to cope with the increasingly prevalent cardiometabolic disease.

Although a compensatory increase in blood insulin levels is required to maintain glucose homeostasis in insulin resistance (IR) individuals[6], sustained hyperinsulinemia in obesity or the early phase of T2D may further impair insulin signal transduction[7,8], resulting in a vicious cycle between hyperinsulinemia and IR, thereby contributing to the development of cardiometabolic disease. Indeed, IR has been recognized as the main cause of metabolic dysfunction and diabetes, but it is presently unclear whether there is a causal relationship between hyperinsulinemia and IR. In addition, the mechanism underlying primary hyperinsulinemia also remains elusive. Exploring the causes of increased insulin production will not only shed light on our understanding of the pathogenesis of cardiometabolic disease, but also reveal novel therapeutic targets for the treatment of these conditions.

Genome-wide association studies have shown many associations between common genetic variations and complex metabolic disorders[9–11]. However, recent efforts with next-generation sequencing technology have highlighted genetic studies of extreme phenotypes as an alternative approach to identify candidate rare variants with large effects underpinning complex

diseases[12–15]. Notably, besides genetic makeup, environmental factors such as differences in diet or medication contribute substantially to the pathogenesis of metabolic diseases, hindering genetic studies in human populations. As a model species closely related to humans, the rhesus monkey presents a unique model to study complex metabolic diseases, due to its human-like genome, the controllability of environmental factors, as well as the feasibility of real-time monitoring of the metabolic phenotypes. Previously, we have established a rhesus monkey model with spontaneous MetS that recaptures most features of the human disease as well as the response to clinically used treatment[16]. In addition, we have developed RhesusBase to provide an information-rich framework for monkey genomics studies[17–19]. These efforts provide an opportunity to identify novel candidate genes that underlie complex metabolic diseases in the genomic context of the rhesus monkey. In this study, through the combination of primate-based extreme phenotype sampling and mice model-based verification, we reveal that mutation of *PRKD2* is involved in the pathogenesis of hyperinsulinemia which, in turn, results in IR and metabolic disorders. The findings not only uncover a novel function of *PRKD2* in regulating insulin secretion but also reveal a potential therapeutic target for metabolic diseases.

## Results

**Identification of monkeys with extremely high insulin levels and IR.** We established a cohort of rhesus monkeys with spontaneous MetS and followed the animals up to 40 months (Fig. 1 and Supplementary Fig. 1). The fasting plasma insulin levels of these monkeys were significantly higher than in age-matched controls (Fig. 1a). Notably, we found two monkeys (ID# as 950807 and 960109) with extremely elevated fasting insulin levels

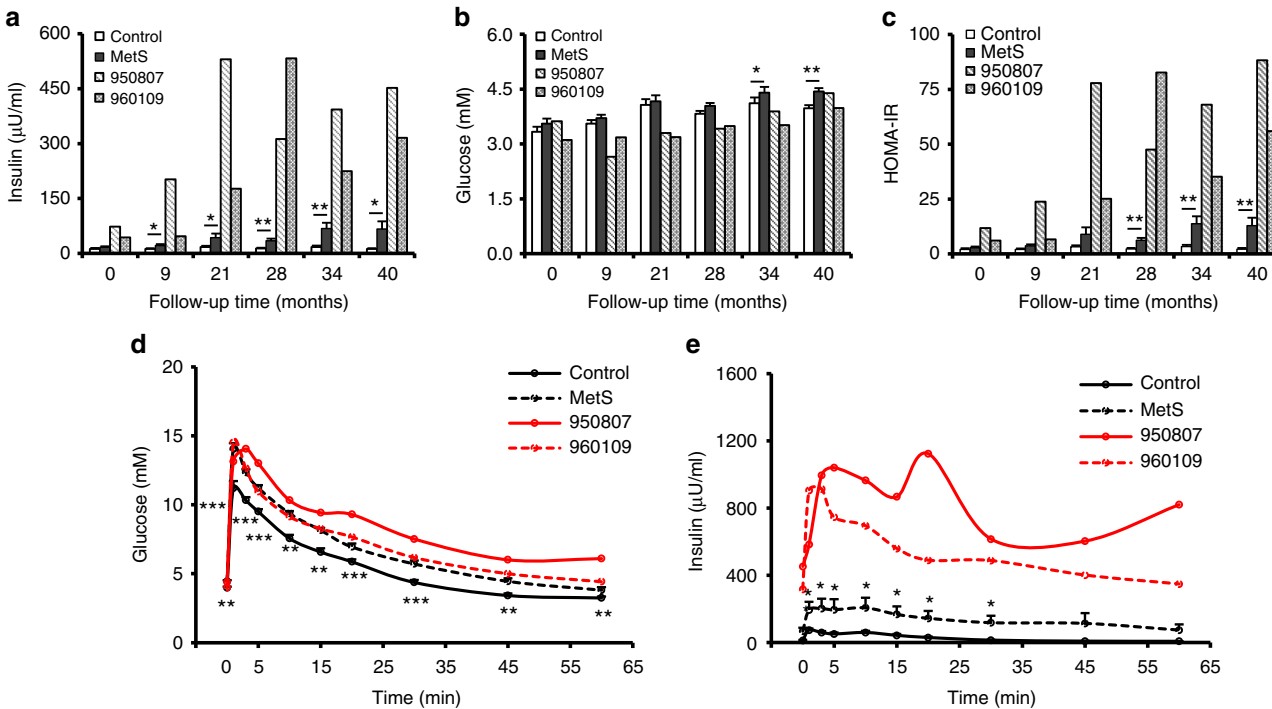

**Fig. 1** Changes of glucose and insulin levels in hyperinsulinemic monkeys. **a** Fasting plasma insulin levels of control, MetS, and hyperinsulinemic monkeys in follow-up studies. **b** Fasting plasma glucose levels of control, MetS, and hyperinsulinemic monkeys in follow-up studies. **c** Homeostasis model assessment of insulin resistance (HOMA-IR) of control, MetS, and hyperinsulinemic monkeys in follow-up studies. **d** Glucose excursion of control, MetS, and hyperinsulinemic monkeys during IVGTT at 40 months of follow-up studies. **e** Insulin excursion of control, MetS, and hyperinsulinemic monkeys during IVGTT at 40 months of follow-up studies. *$p < 0.05$, **$p < 0.01$, ***$p < 0.001$, control vs MetS (Control, $n = 16$; MetS, $n = 17$). All data are represented as mean ± SEM, the significant difference between groups was assessed by the Student's $t$-test

relative to the rest of the MetS group (Fig. 1a). At 40 months, the fasting serum insulin levels were 5- to 7-fold greater than that of the rest of the MetS group (316 μU/ml for 960109, 452 μU/ml for 950807, and 67 μU/ml for MetS) (Fig. 1a). Despite the extreme hyperinsulinemia, the blood glucose levels were only moderately reduced at the early stages and returned to the normal level in the later phase, as compared with the control or the MetS groups (Fig. 1b). Both hyperinsulinemic monkeys developed severe IR, as manifested by profoundly increased HOMA-IR (homeostasis model assessment of IR) compared to the control or MetS animals (Fig. 1c). Although the glucose tolerance tests did not show obvious differences between the hyperinsulinemic monkeys from that of the MetS (Fig. 1d), the basal and glucose-stimulated insulin levels were profoundly greater in these hyperinsulinemic monkeys than that in controls or MetS monkeys (316, 452, and 67 μU/ml for #960109, #950807, and MetS at basal, respectively; and 913, 1124, and 206 μU/ml for 960109, 950807, and MetS after glucose challenge, respectively) (Fig. 1e). Thus, both monkeys with extreme hyperinsulinemia exhibited markedly augmented glucose-induced insulin secretion and severe IR.

**Metabolic phenotypes of hyperinsulinemic monkeys.** As insulin signaling is implicated in multiple metabolic parameters, we next analyzed the metabolic features of these two hyperinsulinemic monkeys (Fig. 2 and Supplementary Fig. 2). Both hyperinsulinemic monkeys, particularly monkey 950807, were more obese, as evidenced by increased body weight and waist size (Fig. 2a, b). Since their waist/hip ratio was similar to that of the MetS group (Fig. 2c), these two monkeys did not show more

severe abdominal obesity. Similar to other MetS monkeys, both the hyperinsulinemic individuals developed systolic rather than diastolic hypertension (Fig. 2d, e). The hyperinsulinemia was associated with further reduced high-density lipoproteins-cholesterol (HDL-c) (Fig. 2f). But surprisingly, it was accompanied by a decreased rather than an increased in triglyceride (TG) levels in both monkeys, in particular # 960109, compared to the MetS group (Fig. 2g). The hypoglycemia and low TG might be attributed to the markedly elevated blood insulin level (Figs. 1b and 2g); and the total cholesterol (TCh) and low-density lipoproteins-cholesterol (LDL-c) also showed a decreasing trend (Fig. 2h, i). However, the long-term glucose metabolic marker, hemoglobin A1c (HbA1c), was higher in hyperinsulinemic monkeys, particularly in #950807, than in the control or the MetS group (Fig. 2j), suggesting late-onset impairment of glucose tolerance and IR in the hyperinsulinemic monkeys. This was consistent with increased HOMA-IR in the hyperinsulinemic monkeys (Fig. 1c). In addition, echocardiography showed an increasing trend in the left ventricular wall and myocardial contractility as evidenced by a greater ejection fraction and fractional shortening in the hyperinsulinemic monkeys at 40 months of follow-up studies (Fig. 2k, l). Cardiac hypertrophy and enhanced contractility of the hyperinsulinemic monkeys might be attributed to myocardial adaptive responses to increased insulin stimulation. Taken together, our data suggest that hyperinsulinemia leads to obesity, hypertension, and IR, featuring MetS, but different from conventional spontaneous MetS in terms of lipid metabolism, as manifested by reduced rather than elevated blood concentrations of TG, TCh, and LDL-c, likely due to high insulin-stimulated uptake of lipids by peripheral tissues.

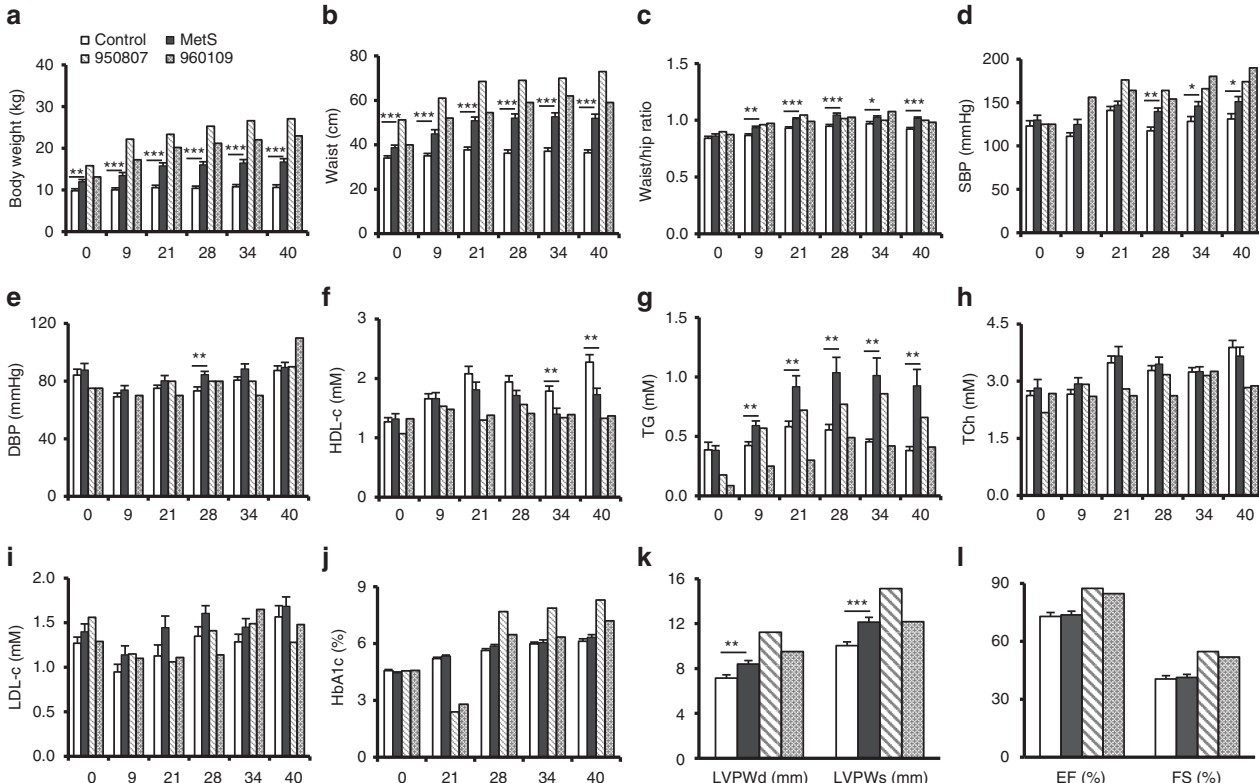

**Fig. 2** Metabolic and anthropometric parameters of control, MetS, and hyperinsulinemic monkeys. **a** Body weights. **b** Waist circumferences. **c** Waist/hip ratios. **d** SBP, systolic blood pressures. **e** DBP, diastolic blood pressures. **f** HDL-c, high-density lipoprotein cholesterol. **g** TG, triglyceride. **h** TCh, total cholesterol. **i** LDL-c, low-density lipoprotein cholesterol. **j** HbA1c, hemoglobin A1c. **k** LVPWd (left ventricular posterior wall thickness at end-diastole) and LVPWs (left ventricular posterior wall thickness at end-systole) at 40 months of follow-up studies. **l** EF (ejection fraction) and FS (fraction shortening) at 40 months of follow-up studies. *$p < 0.05$, **$p < 0.01$, ***$p < 0.001$, control vs MetS (Control, $n = 16$; MetS, $n = 17$). All data are represented as mean ± SEM, the significant difference between groups was assessed by the Student's $t$-test

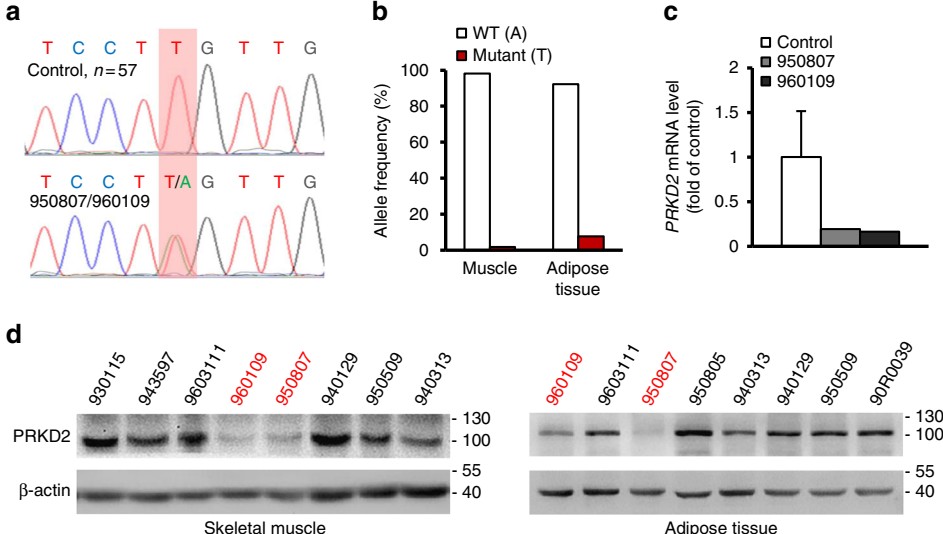

**Fig. 3** *PRKD2* K410X variation is shared by two hyperinsulinemic monkeys. **a** Sanger sequencing indicating one heterozygous nonsense variation in *PRKD2* (K410X) exclusive to the two cases (950807, 960109). **b** As for the *PRKD2* nonsense variation, the allele frequencies of the two alleles (WT, Mutant) were shown, according to the RNA-seq data in muscle and adipose tissue from a *PRKD2* mutant monkey (950807). **c** Real-time PCR quantifications of the *PRKD2* expression in skeletal muscle of WT and the *PRKD2* mutant monkeys. **d** Western blots of *PRKD2* protein expression in skeletal muscle (left) and adipose tissues (right) from control (black) and hyperinsulinemic (red) monkeys

**PRKD2 K410X mutation was identified in hyperinsulinemic monkeys.** As all these animals were living in the same accredited animal facility, were fed the same monkey chow diet, and experienced well-controlled environmental conditions, it is likely that some case-specific genetic variants may underlie the extreme phenotype in these two monkeys. We thus performed genome-wide, exon-centric sequencing of the two monkeys and five normal controls from our rhesus monkey cohort to identify candidate genetic variations potentially linked with the hyper-insulinemic phenotype. DNA fragments in exonic regions were captured and sequenced at ~100× coverage on an Illumina sequencer, and an average of ~90% of the coding regions in rhesus monkey were successfully captured and sequenced (Supplementary Table 1 and Supplementary Fig. 3; Methods). On the basis of the deep sequencing data, we then set out to identify case-specific genetic variations. Notably, considering these animals were selected from an in-house cohort population of rhesus monkeys with a similar genetic background, the number of candidate variations should be low, as most unrelated alleles would have been removed. As expected, only six variations with large effects (three frameshift variations, two nonsenses variations, and one variation introducing an aberrant splice site), as well as 163 missense variations were found to be exclusive to the two hyperinsulinemic monkeys (Supplementary Table 2; Methods). By further taking advantage of the whole-genome sequencing data on 31 normal monkeys archived in RhesusBase[17,18], only one heterozygous nonsense variation (K410X) in serine/threonine protein kinase D2 (*PRKD2*), and 28 missense variations were identified to be exclusive to the two cases (Supplementary Table 2; Methods).

Considering that the nonsense variations have typically stronger effects on the proteome by directly obstructing the open reading frames of proteins, we focused on the *PRKD2* nonsense variation (K410X) in the subsequent experimental verifications with Sanger sequencing in the two cases, as well as in the in-house cohort population of 57 monkeys without extremely high insulin levels. Notably, the heterozygous nonsense variation was found to be exclusive to the two cases (Fig. 3a), suggesting its potential roles in the regulation of the extreme phenotype.

The K410 site is highly conserved across species ranging from zebrafish to humans (Supplementary Fig. 4). The K410X mutation probably blocked the RNA and protein expression of *PRKD*2 via nonsense-mediated mRNA decay, as the mRNA with mutant alleles was significantly degraded, leading to differentially distributed RNA-Seq reads between WT and mutant alleles ($p = 0.01$, Fisher's exact test, Fig. 3b) and the mRNA levels of *PRKD2* were lower in the mutant (hyperinsulinemic monkey) than in the WT monkeys (Fig. 3c). Matching the decreased level of mRNA expression, the protein expression of *PRKD*2 was also significantly down-regulated in cases with the K410X nonsense mutation (Fig. 3d). Animals with such a mutation should thus exhibit an effect similar to *PRKD2* knockdown.

**PRKD2 deficiency leads to increased insulin levels and IR in mice.** The identification of *PRKD2* nonsense mutation in hyperinsulinemic monkeys further motivated us to investigate the relationship between down-regulation of *PRKD2* and hyper-insulinemia in *PRKD2* gene knockout ($PRKD2^{-/-}$) mouse model (Supplementary Fig. 5a–c). When quantifying the insulin levels and activities of the insulin signaling in 4-week-old mice, we found an increased fasting insulin level and decreased fasting glucose level in $PRKD2^{-/-}$ mice when compared with WT mice (Supplementary Fig. 6a, b), but the body weight and insulin-stimulated Akt phosphorylation in skeletal muscle and liver were comparable between the two groups (Supplementary Fig 6c–g).

At 14 weeks, both the basal and glucose-stimulated insulin secretion were greater in $PRKD2^{-/-}$ than in WT mice (Fig. 4a, b). The $PRKD2^{-/-}$ mice also exhibited slightly improved glucose tolerance as demonstrated by intraperitoneal glucose tolerance tests and the area under curve (AUC) of glucose (Fig. 4c, d), possibly due to the hyper-secretion of insulin in $PRKD2^{-/-}$ mice. Further studies with high-fat diet challenge may provide additional information to interpret phenotypes detected in $PRKD2^{-/-}$ mice. Besides the increased insulin level, blood lipids were comparable in the $PRKD2^{-/-}$ and WT mice (Supplementary Fig. 7a), the body weight of $PRKD2^{-/-}$ mice was significantly increased (Supplementary Fig. 7b), and the insulin-stimulated

Akt ser473 phosphorylation in liver (Fig. 4e, f) or skeletal muscle (Fig. 4g, h) was markedly decreased in $PRKD2^{-/-}$ compared with WT mice, indicating the development of obesity and IR. We further performed in vivo insulin tolerance test (ITT) to verify the existence of IR in $PRKD2^{-/-}$ mice, and found that the blood glucose level was significantly higher in $PRKD2^{-/-}$ mice than that in WT controls after insulin injection (Fig. 4i, j), supporting the existence of IR in $PRKD2^{-/-}$ mice. These findings were

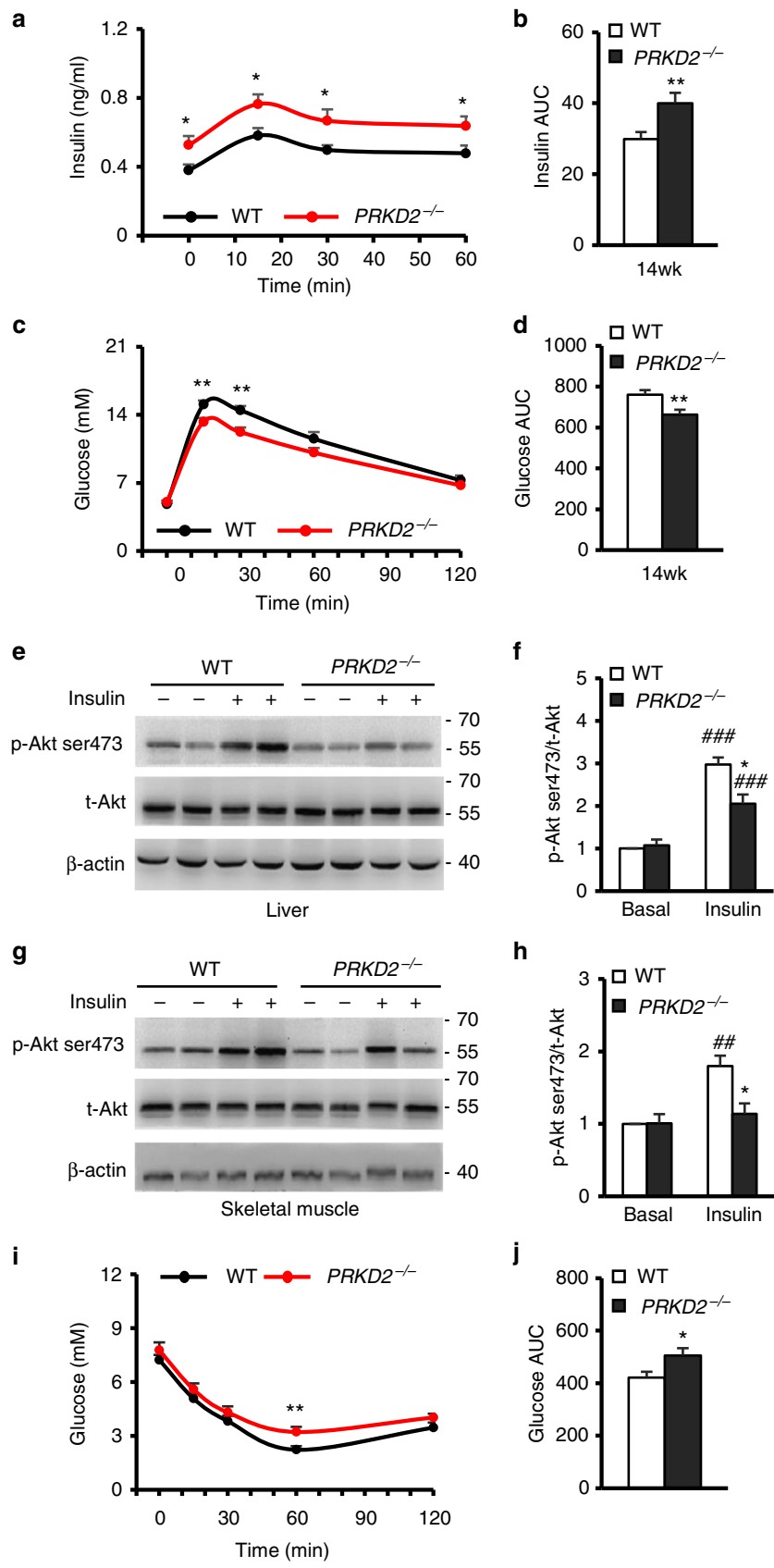

qualitatively similar to those of the PRKD2 mutant monkeys, but the overall phenotype in the mice was milder, likely due to species differences. All together, these results strongly suggest that down-regulation of PRKD2 expression is sufficient to increase insulin level, and support a hyperinsulinemia-primary model that IR and obese progressively developed following hyperinsulinemia.

**PRKD2 deficiency increases insulin secretion**. Previous studies have suggested that high insulin levels are associated with increased β-cell mass, β-cell secretion, or IR[20–22]. To delineate the mechanism underlying PRKD2 deficiency-induced hyper-insulinemia, we analyzed the islet morphology and found no overt difference between PRKD2[−/−] and WT mice in terms of islet size or the proportion of β-cells within islets (Fig. 5a–d), indicating that PRKD2 deficiency does not affect islet structure.

We next investigated insulin secretion in isolated perfused islets from both PRKD2[−/−] and WT mice and found that insulin secretion in response to high glucose in islets from PRKD2[−/−] mice was significantly higher than that in WT islets (Fig. 5e). Because islets contain several types of cells, and hormones secreted from these cells may regulate β-cell secretion, we performed PRKD2 gene silencing with siRNA in cultured Ins-1 cells, a β-cell line, to investigate the sequelae of PRKD2 deficiency. Indeed, knocking-down of PRKD2 enhanced both basal and high glucose-induced insulin secretion (Fig. 5f). Moreover, we found that PRKD2 deficiency did not alter insulin-stimulated Akt phosphorylation in Ins-1 cells (Fig. 5g, h), suggesting that PRKD2 deficiency-induced insulin secretion was not mediated by enhanced insulin signaling. Taken together, these results jointly indicate that a deficiency in PRKD2 leads to hyperinsulinemia by regulating insulin secretion rather than altering insulin signaling.

It has been shown that $Ca^{2+}$ regulates a sequence of core events of insulin exocytosis, including vesicle docking and fusion, and the activity of the secretory machinery[23–25]. Next, we sought to determine whether PRKD2 deficiency affects intracellular $Ca^{2+}$ concentration ($[Ca^{2+}]_i$) in response to high glucose stimulation in islets isolated from PRKD2[−/−] and WT mice. A $Ca^{2+}$ indicator (Fluo-4 AM) was used to monitor the changes of $[Ca^{2+}]_i$. Increasing glucose concentration from 2.8 to 20 mM in the perfusion solution induced a robust global $Ca^{2+}$ elevation (Fig. 6a, b), consistent with previous reports[26]. Notably, the high glucose-evoked increase in $[Ca^{2+}]_i$ was significantly augmented in islets from PRKD2[−/−] mice ($F/F_0$, 2.66 ± 0.2) relative to their WT counterparts ($F/F_0$, 2.08 ± 0.16) (Fig. 6b). In addition, PRKD2 deficiency was able to augment $Ca^{2+}$ influx in response to membrane depolarization induced by high KCl (Fig. 6a, b).

Indeed, using patch-clamp analyses, we found that nifedipine-sensitive L-type $Ca^{2+}$ current was significantly greater in β-cells from PRKD2[−/−] mice (Fig. 6c, d). In addition, we performed real-time PCR analysis to qualify the mRNA level of CACNA1c in isolated mouse β-cells and found a significantly increased expression in PRKD2[−/−] mice when compared to WT mice (Fig. 6e). When treated the β-cells with nifedipine, an L-type $Ca^{2+}$ channel blocker, the differences in $Ca^{2+}$ current between PRKD2[−/−] and WT mouse β-cells became undetectable (Fig. 6c).

Notably, when treated monkeys with nifedipine during intravenous glucose tolerance tests (IVGTTs), we found that low-dose intravenous nifedipine injection markedly inhibited insulin secretion, especially glucose-stimulated insulin secretion, in PRKD2 mutant monkey (Fig. 6f). In contrast, only mild inhibition in glucose-stimulated insulin secretion was detected in control monkey (Fig. 6f), possibly due to the low basal insulin secretion. The glucose excursion was not affected in both mutant and control monkeys (Fig. 6g). These lines of evidence thus suggest that enhanced L-type $Ca^{2+}$ channel expression and activity are essentially involved in PRKD2 deficiency-induced increase of insulin secretion.

## Discussion

In this study, we have provided multiple lines of evidence that PRKD2 down-regulation is sufficient to enhance β-cell insulin secretion and subsequently induce hyperinsulinemia and systemic IR. First, using extreme phenotype sampling in conjunction with deep sequencing analysis, we have identified a nonsense mutation of PRKD2 (K410X) which is shared by both extreme hyper-insulinemia cases but not the other 57 rhesus monkeys of the same cohort. Second, the PRKD2 knockout caused hyper-insulinemia in mice. Mechanistically, we have demonstrated that PRKD2 deficiency sensitizes glucose-induced insulin secretion via increasing $Ca^{2+}$ influx by induction of L-type $Ca^{2+}$ channel expression and activity. Moreover, intravenous injection of nife-dipine, an L-type $Ca^{2+}$ channel blocker, markedly inhibited basal and the glucose-stimulated high insulin secretion in PRKD2 mutant monkey. These findings thus not only uncover a novel function of PRKD2 in regulating β-cell insulin secretion but also reveal a potentially important therapeutic target for metabolic diseases. Although the human orthologous site of the monkey nonsense mutation is not polymorphic according to the 1000 Genomes Project, it is plausible that patients with loss-of-function mutations on PRKD2 gene should have similar phenotype of hyperinsulinism.

PRKD family belongs to the calmodulin-dependent protein kinase superfamily, which contains three members, PRKD1 (PRKDμ), PRKD2, and PRKD3 (PRKDv)[27–29]. Previous studies have revealed that these PRKD isoforms are expressed differentially in various exocrine and endocrine cells of the pancreas. Specifically, PRKD1 and PRKD2 are more abundant in the islets of the mouse and human pancreas, whereas PRKD3 predominates in exocrine cells[30]. Although PRKD has been linked to cardio-myocyte $Ca^{2+}$ handling[31], it remains unknown whether PRKD could modulate β-cell L-type $Ca^{2+}$ channel activity, and play some roles in the regulation of insulin secretion. The findings here thus pinpoint a novel function of PRKD2 in regulating β-cell insulin secretion in that PRKD2 deficiency primarily increases insulin secretion, and further induces obesity and IR. Although our data support the hyperinsulinemia-causes-IR hypothesis, short-term PRKD2 inhibition or deficiency may have beneficial effect via enhancing glucose-stimulated insulin secretion. Nevertheless, sustained PRKD2 suppression or deficiency is detrimental.

**Fig. 4** PRKD2 deficiency enhanced Insulin secretion and lead to IR in mice. **a**, **b** Insulin excursion during IPGTTs (**a**) and area under the curve of insulin (**b**) in mice with the indicated genotypes at 14 weeks of age. *$p < 0.05$, **$p < 0.01$, WT vs PRKD2[−/−] (WT, $n = 24$; PRKD2[−/−], $n = 27$). **c**, **d** Glucose excursion during IPGTTs (**c**) and area under the curve of glucose (**d**) in mice with the indicated genotypes at 14 weeks of age. **$p < 0.01$, WT vs PRKD2[−/−] (WT, $n = 22$; PRKD2[−/−], $n = 28$). **e**, **f** Western blots of total Akt and Akt phosphorylation at ser473 in liver from WT and PRKD2[−/−] mice at age of 14 weeks old. *$p < 0.05$, WT vs PRKD2[−/−]; ###$p < 0.001$, basal vs insulin-treatment ($n = 8$/group). **g**, **h** Western blots of total Akt and Akt phosphorylation at ser473 in skeletal muscle from WT and PRKD2[−/−] mice at the age of 14 weeks old. *$p < 0.05$, WT vs PRKD2[−/−]; ##$p < 0.01$, basal vs insulin-treatment ($n = 8$/group). **i**, **j** Glucose excursion during ITTs (**i**) and area under the curve of glucose (**j**) in mice with the indicated genotypes at 14 weeks of age (WT, $n = 16$; PRKD2[−/−], $n = 17$). *$p < 0.05$, **$p < 0.01$, WT vs PRKD2[−/−]. All data are represented as mean ± SEM, the significant difference between groups was assessed by the Student's t-test

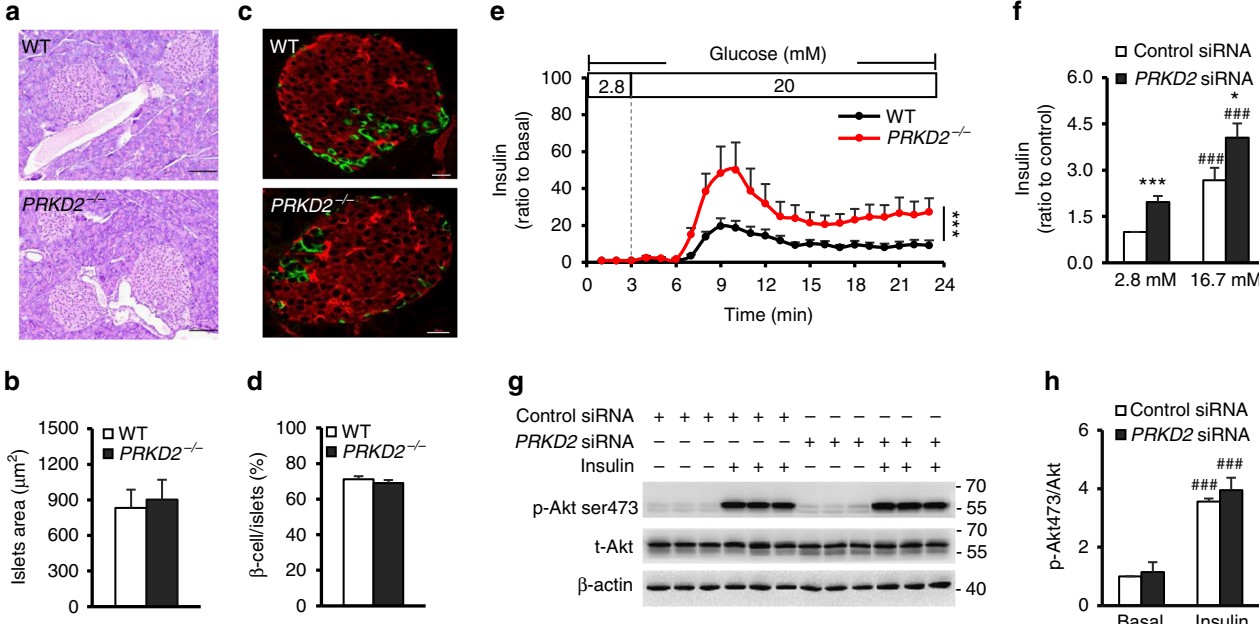

**Fig. 5** *PRKD*2 deficiency increases insulin secretion in isolated islets and β cells. **a, b** Representative pancreatic images of hematoxylin and eosin (HE) staining (**a**) and statistics (**b**) showing that *PRKD2*-knockout in mice did not affect islet morphology and size at 14 weeks of age. Scale bar = 100 μm (WT, *n* = 8; *PRKD2*$^{-/-}$, *n* = 6). **c, d** Representative images of immunofluorescence staining for insulin and glucagon in pancreatic islets (**c**) and statistics (**d**) showing that *PRKD2*-knockout did not affect the number of β-cells within islets. Scale bar = 20 μm (WT, *n* = 6; *PRKD2*$^{-/-}$, *n* = 5). **e** Time-courses of glucose-stimulated insulin secretion in islets isolated from *PRKD2*$^{-/-}$ (red) and WT (black) mice. ***$p < 0.001$, WT vs *PRKD2*$^{-/-}$, by two-way ANOVA (*n* = 10/group). **f** Insulin secretion in control and *PRKD2*-knockdown Ins-1 cells. *$p < 0.05$, ***$p < 0.001$, control siRNA vs *PRKD2* siRNA; ###$p < 0.001$, 2.8 vs 16.7 mM glucose (*n* = 10/group). **g, h** Representative western blots of Akt ser473 phosphorylation (**g**) and statistics (**h**) in control and *PRKD2*-knockdown Ins-1 cells. ###$p < 0.001$, basal vs insulin (*n* = 8/group). All data are represented as mean ± SEM, the significant difference between groups was assessed by the Student's *t*-test

Although the metabolic syndrome was considered as a "common disease" stemming from the additive effects of common variants, some rare variations with large effects were also proven to account for the phenotypes. As a proof-of-concept study with two monkey animals showing extreme hyperinsulinemia, we tested whether such rare variations with large effects do exist underpinning the extreme phenotype. Such a simplified monogenic model led to the identification of one nonsense mutation on *PRKD2* genes as a candidate. However, the polygenic inheritance model, as well as the contributions of other missense mutations with relatively milder effects on the proteome, could not be ruled out and may deserve future investigations. Specifically, knock-in monkeys with this mutation would be the most straightforward means to verify the causality between the *PRKD2* nonsense mutation and the extreme metabolic phenotype, but remain an expensive, time consuming and technically challenging option in monkeys. However, such a proof-of-concept study in rhesus monkeys prompted us to investigate the metabolic phenotypes of this previously neglected candidate gene in mouse models. Overall, although more macaque animals, as well as knock-in monkeys, are needed to verify the causality with statistical confidence, the study here suggests a practical approach to identify disease-related genes through the combination of primate-based extreme phenotype sampling and mice model-based verification.

Rhesus monkeys share high genetic homology and functional similarities with humans[16,32–35], and thus provide ideal models for studying human diseases. Using integrative approaches, we identified a *PRKD2* nonsense mutation in rhesus monkeys with hyperinsulinemia. In addition, as a cluster of metabolic disorder, IR, overweight, hypertension, decreased HDL-c, and ventricular hypertrophy were found in these cases, compared with control monkeys and the MetS cohort. Given that insulin is a known

mitogen that regulates protein synthesis and neoplasia, enhance plasma insulin binds to and activates insulin receptors/insulin-like growth factor-1 receptors and the downstream insulin receptor substrate/phosphatidylinositide 3-kinases axis, promoting peripheral tissue growth and weight gain[36–38]. In addition, previous studies have shown that hyperinsulinemia is associated with hypertension and dyslipidemia[39,40], likely due to insulin-mediated sympathetic stimulation and IR in peripheral tissues[41,42]. It is also noteworthy that the TG and TCh were moderately reduced compared with the control and the MetS groups, revealing a unique lipid profile that differs from what is known in the general population of humans with MetS. Although it remains technically challenging to generate large populations of transgenic or knockout disease models in primates[43–46], the manipulation of important candidate genes in primates is needed in future studies to fully clarify the pathogenesis of complex metabolic diseases.

It remains controversial as to whether hyperinsulinemia causes IR and MetS[3–7]. Upregulated insulin secretion facilitates glucose uptake in peripheral tissues and reduces blood glucose[47], especially in the absence of IR. However, a persistent increase in blood insulin may cause IR[48]. Our in vitro work with siRNA to knock down *PRKD2* in Ins-1 cells and Hepa1-6 cells (Supplementary Fig. 8), as well as the quantifications of insulin levels and activities of the insulin signaling in 4-week- and 14-week-old *PRKD2*$^{-/-}$ mice (Fig. 4 and Supplementary Fig. 6) suggests a hyperinsulinemia-primary model that the development of obesity and IR following hyperinsulinemia. Peripheral IR may then exaggerate hyperinsulinemia, resulting in a vicious cycle between hyperinsulinemia and systemic IR.

To delineate the molecular mechanism underlying *PRKD2* deficiency-induced increase in insulin level, we assessed insulin

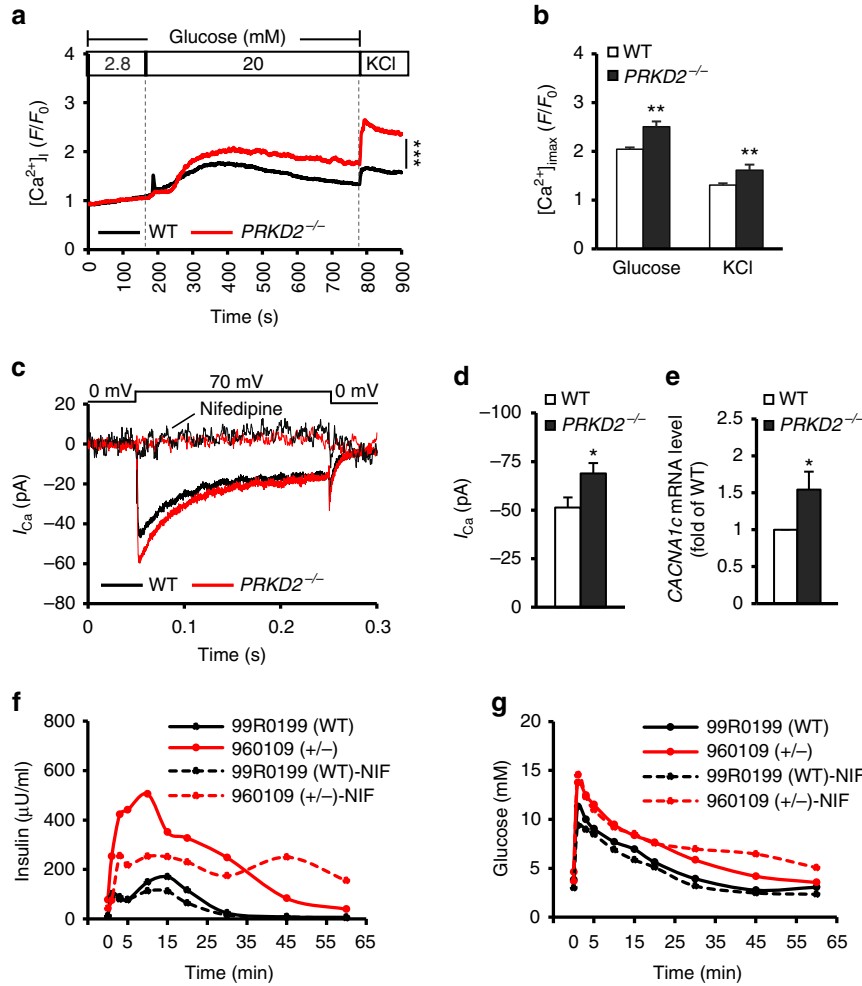

**Fig. 6** *PRKD2* deficiency induces insulin secretion through regulation of $Ca^{2+}$ influx in β-cells. **a** Time-course of $[Ca^{2+}]_i$ induced by 20 mM glucose in *PRKD2*$^{-/-}$ (red) and WT β-cells (black), *x*-axis indicates the time. \*\*\**p* < 0.001, WT vs *PRKD2*$^{-/-}$, by two-way ANOVA. **b** Peak levels of $[Ca^{2+}]_i$ in response to glucose or high KCl in WT and *PRKD2*$^{-/-}$ β-cells. \*\**p* < 0.01, WT vs *PRKD2*$^{-/-}$ (representative data of five experiments; WT, *n* = 5; *PRKD2*$^{-/-}$, *n* = 7, and *n* > 9 cells/group). **c** Time-course of $Ca^{2+}$ currents induced by membrane depolarization with or without nifedipine in *PRKD2*$^{-/-}$ (red) and WT β-cells (black), *x*-axis indicates the time (WT, *n* = 16 β-cells/5 mice; *PRKD2*$^{-/-}$, 14 β-cells/6 mice). **d** Peak levels of $Ca^{2+}$ currents in depolarization in WT and *PRKD2*$^{-/-}$ β-cells. \**p* < 0.05, WT vs *PRKD2*$^{-/-}$ (WT, *n* = 16 β-cells/5 mice; *PRKD2*$^{-/-}$, *n* = 14 β-cells/6 mice). **e** mRNA expression levels of *CACNA1c* in *PRKD2*$^{-/-}$ and WT β-cells. \**p* < 0.05, WT vs *PRKD2*$^{-/-}$ (WT, *n* = 8, *PRKD2*$^{-/-}$, *n* = 6). **f** Basal and glucose-stimulated insulin secretion in *PRKD2* mutant and WT rhesus monkey in IVGTT with or without nifedipine. **g** Glucose excursion in *PRKD2* mutant and WT monkeys during IVGTT with or without nifedipine. All data are represented as mean ± SEM; the significant difference between groups was assessed by the Student's *t*-test

secretion in *PRKD2*-deficient islets and cultured β-cells. We have shown that down-regulation of *PRKD2* promotes insulin secretion through increasing $Ca^{2+}$ influx that is dependent on L-type $Ca^{2+}$ channel expression and activity. Regarding a potential role of protein kinase D in regulating $Ca^{2+}$ signaling, previous studies have shown that activation of the gene increases the phosphorylation of cardiac myosin-binding protein C at Ser315, sensitizing myocardial responsiveness to $Ca^{2+}$, resulting in a positive inotropic effect[49]. Here, we have shown that *PRKD2* may suppress glucose-induced increase in cytoplasmic $Ca^{2+}$, since its down-regulation markedly augments the elevation of $[Ca^{2+}]_i$ induced by high glucose and membrane depolarization. In addition, we found *PRKD2* deficiency can increase the expression and activity of L-type $Ca^{2+}$ channel in *PRKD2*$^{-/-}$ β-cells, further strengthening that *PRKD2* deficiency enhances insulin secretion by regulate β-cell $Ca^{2+}$ influx.

## Methods

**Rhesus monkeys.** The Rhesus monkey MetS model has been reported previously[16]. All monkeys were housed individually in cages, had free access to water,

and were fed ad libitum with pellet monkey chow (Beijing HFK Bio-Technology Co., Ltd, China) that contained 7–10% crude fat, 16–20% crude protein, and 55–65% crude carbohydrate. All animals in this study were maintained in an animal facility at Peking University that is accredited by the Association for Assessment and Accreditation of Laboratory Animal Care (AAALAC). The experimental procedures for monkeys were approved by the Institutional Animal Care and Use Committee of Peking University and were in accordance with the principles of laboratory animal care of the National Academy of Sciences/National Research Council (approval number: IMM-ZhangXQ-1).

**Blood sampling and biochemical tests.** The blood samples were taken from a hindlimb vein after overnight fasting and anesthesia with ketamine at 10 mg/kg body weight. Blood samples for HbA1c analysis were collected in vacutainer tubes containing ethylenediaminetetraacetic acid (EDTA). For plasma samples, blood was collected in vacutainer tubes containing EDTA, mixed and kept on ice, and centrifuged for 10 min at 1200 × *g* at 4 °C within an hour. All measurements of plasma lipids and glucose were performed using Roche biochemical test kits. Insulin was measured by Roche Modular Analytics E170 Combinations.

Intravenous glucose tolerance test: After 14–16 h overnight fasting, under ketamine anesthesia, two i.v. cannulae were inserted into veins of both hindlimbs, one for blood sampling and the other for glucose administration. After baseline blood sampling, 50% glucose was administered at a dose of 0.5 ml/kg body weight over a time period of 30 s, which made a total glucose load of 0.25 g/kg body weight. Then blood samples were taken at 1, 3, 5, 10, 15, 20, 30, 45, and 60 min

after the challenge. Heparinized saline was infused between each sampling to maintain the patency of the i.v. cannula. All blood samples were immediately put on ice and centrifuged at 4 °C as for plasma isolation. IR was calculated by means of the homeostasis model assessment–IR (HOMA-IR): fasting plasma glucose × fasting insulin/22.5.

**Generation of global *PRKD2*-deficient mice**. The mouse *PRKD2* DNA clones were isolated from a 129SVJ mouse genomic library (Stratagene). A vector that contains a neomycin selection cassette flanked by FRT sites was used to generate *PRKD2* targeting construct. Briefly, two homology arms of *PRKD2* gene were cloned into the targeting vector, and a 603 bp fragment containing exon2 of *PRKD2* was inserted after the first LoxP site (Supplementary Fig. 5). *PRKD2* targeting vector was then linearized with *Not*I, and subsequently electroporated into R1 embryonic stem (ES) cells. G418-resistant ES clones were screened for homologous recombination by DNA blot analysis, as described below. Two independent homologous recombinant ES clones for each targeting construct were micro-injected into blastocysts from C57BL/6J mice to generate male chimeras. Male chimeras were bred with female Black Swiss mice to generate germline transmitted floxed heterozygous mice (*PRKD2*[+/flox-neo]), which was subsequently crossed with Pro-Cre mouse[50] to generate heterozygous mice (*PRKD2*[+/−]). Offspring from intercrosses were genotyped by PCR analysis using mouse tail DNA as previously described[51]. The experimental procedures for mice were approved by the Institutional Animal Care and Use Committee of Peking University and were in accordance with the principles of laboratory animal care of the National Academy of Sciences/National Research Council (Approval number: IMM-ZhangXQ-6).

**DNA analysis for genotyping**. Genomic DNA was extracted from G418-resistant ES cell clones and mouse tails, as previously described[52]. ES cell DNA was digested using Acc65I, and DNA was electrophoresed on a 1% (wt/vol) agarose gel, and subsequently blotted onto a nitrocellulose membrane. A 250 bp fragment was generated by PCR using mouse genomic DNA and was subsequently radiolabeled using [$^{32}$P] dATP by random priming (Invitrogen). DNA blots were hybridized with the radiolabeled probe and visualized by autoradiography. Offspring from intercrosses were genotyped by PCR analysis using mouse tail DNA and the following primers: *PRKD2*, WT/flox allele (forward, AGAGCCAGGTAACAGGAA-CAATAG; reverse, GTGCTAAGGAGGGAGGCTCT), mutant allele (neo-specific primer, AATGGGCTGACCGCTTCCTCGT; reverse, GCAAGCTACTTCCTCCCAAG).

After genotyping, the mice were backcrossed to C57/BL6 for seven generations to reduce contamination from its original mixture with the 129SvJ background. The mice were maintained under specific pathogen-free conditions in an animal facility at Peking University that is accredited by the Association for Assessment and Accreditation of Laboratory Animal Care (AAALAC). The mice had free access to water and food; 4- and 14-week-old male mice were used in the following experiment. All experimental procedures with animals were approved by the Institutional Animal Care and Use Committee of Peking University and were in accord with the principles of laboratory animal care of the National Academy of Sciences/National Research Council.

**Library preparation and deep sequencing**. Genomic DNA was obtained from blood samples of the monkey animals to prepare libraries for parallel exome capture and deep sequencing[53]. DNA fragments in exonic regions of rhesus monkey were captured and enriched following the manufacturer's protocol (Agilent Technologies). Strand-specific RNA-Seq libraries were prepared from skeletal muscle or adipose tissue derived from rhesus monkey as previously reported[54,55]. Both RNA-Seq and Exome-Seq were performed on a HiSeq 2000 Sequencing System. The deep sequencing data were evaluated following the guidelines of the RhesusBase Quality Score system[19]. Exome-seq data and RNA-Seq data were then mapped with BWA (v0.7.10–r789) and Tophat (v2.0.8), respectively. Only uniquely mapped reads were subjected to subsequent analyses. All the raw sequencing data are available at SRA under accession numbers SRP070922 and SRP071778.

**Identification of candidate genetic variations in rhesus monkeys**. In past years, we developed a rhesus monkey MetS model and developed a strategy to identify candidate genetic variations underpinning metabolic diseases through extreme phenotype sampling in 59 rhesus monkeys. Within these monkeys, 20 developed spontaneous MetS, two of which (ID# 950807 and 960109) showed prolonged and extremely high fasting insulin levels.

On the basis of the deep sequencing data, genetic variation calling was performed with the GATK Unified Genotyper (v2.7–0.4). Variations with unusual clustering (≥5 variants within a 25-bp window) or located in tandem repeats were removed, and only variations located in coding regions or splice sites were retained. The effects of these variations were then annotated by snpEff (v3.5a). A three-step pipeline was then used to identify candidate genetic variations: (i) the variations should be specific to the two monkeys with the extreme phenotype, but not in normal controls; (ii) the variations with large effects (such as nonsense variation, frameshift or splice-site variation), as well as missense variations were retained; and (iii) the variations should not be identified in the whole-genome sequencing data on 31 normal monkeys[17].

**Mouse glucose and ITT**. For intraperitoneal glucose tolerance test, WT and *PRKD2*[−/−] mice were fasted overnight, and glucose was injected intraperitoneally at 2 g/kg body weight. Blood samples were taken from a tail vein at the indicated time points for glucose and insulin measurement. Glucose levels were measured on a portable glucometer (Roche, ACCU-CHEK® Performa); insulin levels in serum were measured using an ELISA kit (Millipore).

For ITT, WT, and *PRKD2*[−/−] mice were fasted for 4 h, and insulin (Humulin-R, LILLY) was injected intraperitoneally at 1 U/kg body weight. Blood samples were taken from tail vein at 0, 15, 30, 60, and 120 min, and levels of blood glucose were measured.

**Mouse tissue sample collection**. At 4 and 14 weeks of age, the male mice were anesthetized with sodium pentobarbital (100 mg/kg body weight) and sacrificed by cervical dislocation. Blood samples were collected to isolate serum. Samples of skeletal muscle, liver, and pancreas were harvested for further analysis. In brief, the tissues were snap-frozen in liquid nitrogen to isolate protein and mRNA. For histological and immunohistochemically analyses, tissues were fixed in ice-cold 4% paraformaldehyde (PFA) overnight.

**Islet and single β-cell isolation**. To measure in vitro insulin secretion, the pancreases from 14-week-old mice were exposed by a ventral midline incision and perfused with collagenase P (0.5 mg/ml) through the common bile duct. Then the pancreas was dissected and incubated in collagenase P solution at 37 °C for 24 min. Cooled Hanks buffer was added to the incubation mixture to terminate digestion and separate the pancreatic tissue. Then 100–200 islets were picked out by a pipette under a microscope, and cultured in RPMI1640 (7 mM glucose) at 37 °C overnight in a cell culture incubator. Eighty islets with similar size were pre-incubated for 2 h at 37 °C in fresh RPMI1640, then incubated in a modified Krebs–Ringer bicarbonate buffer (KRBB, pH 7.4) that contained 130 mM NaCl, 2.5 mM KCl, 1 mM MgCl₂, 2 mM CaCl₂, 2.8 mM glucose, and 10 mM HEPES, supplemented with 0.1% bovine serum albumin for 30 min to synchronize cells. After pre-incubation, the islets were transferred into perfusion buffer containing 2.8 mM glucose, perfused for 3 min, then was changed to perfusion buffer containing 16.7 mM glucose, and perfused for 20 min at 37 °C. The perfusion rate was 500 μl/min and the perfusate was collected to measure insulin.

To obtain single β-cells, isolated mouse islets were pretreated with Hanks buffer without Ca²⁺/Mg²⁺ at 37 °C for 4 min, and digested into single cells by trypsin for 4 min at 37 °C. Then the separated β-cells were cultured in RPMI1640 containing 10% fetal bovine serum (FBS) and 7 mM glucose for overnight.

**Calcium imaging and patch-clamp analysis in β-cells**. Fluo-4 AM was from Thermo Fisher (Cat#: F14217). For Ca²⁺ imaging experiments, β-cells were incubated with 5 μM Fluo-4 AM in KRBB containing 2.8 mM glucose at 37 °C for 20 min. The cells were perfused with KRBB containing 2.8 mM glucose for 3 min, then switched to KRBB containing 20 mM glucose for 10 min. The depolarization was evoked by high-K⁺ solution (pH 7.4) containing 70 mM KCl, 67 mM NaCl, 1 mM MgCl₂, 2 mM CaCl₂, 10 mM HEPES, and 2.8 mM glucose. Dynamic images were captured on a Zeiss LSM710 confocal microscope with a ×40, 1.3 NA oil-immersion objective, with a frequency of 2 Hz. Data were analyzed using ImageJ (NIH, Bethesda, MD).

To investigate L-type Ca²⁺ currents, patch-clamp was conducted with standard whole-cell recordings using an EPC-10 patch-clamp amplifier (HEKA) at room temperature (~22 °C). Isolated islet cells were stimulated with a train of depolarizing pulses from a holding potential of −70 mV. Cells that exhibited no voltage-dependent Na+ currents were β-cells due to the inactivation of Na+ channel at the −70 mV holding potential[56]. The extracellular solution containing 138 mM NaCl, 5.6 mM KCl, 2.6 mM CaCl₂, 1.2 mM MgCl₂, 5 mM glucose, and 10 mM HEPES (pH 7.4 adjusted with NaOH), while the pipette solution containing 125 mM Cs-glutamate, 0.3 mM Na₂-GTP, 2 mM Mg-ATP, 1 mM MgCl₂, and 0.1 mM EGTA (pH 7.2 with CsOH). For depolarization experiments, Ca²⁺ currents were evoked by a depolarization from −70 mV to 0 mV for 200 ms. For L-type Ca²⁺ channel blocking experiments, nifedipine with a final concentration of 30 μM was added in the bath solution during depolarization[57].

**Cell culture**. The rat β-cell line (Ins-1) was from the China Infrastructure of Cell Line Resources (Beijing, China). Cells were cultured at 37 °C under 5% CO₂ and 95% O₂ in RPMI1640 with 11.1 mM glucose and supplemented with 10% FBS, 1 mM pyruvate, 50 μM β-mercaptoethanol, 100 U penicillin/ml, and 0.1 mg streptomycin/ml. Lipofectamine RNAiMAX from Invitrogen and *PRKD2* siRNA from Dharmacon (Cat#: L-081927-02-0005) were used to knock down *PRKD2* when the cells had reached 60–70% confluence. Insulin in the supernatant was measured after the cells were incubated in KRBB containing 2.8 or 16.7 mM glucose for 30 min. For insulin sensitivity, Ins-1 cells were stimulated with 100 nM insulin for 20 min and proteins were extracted for western blot analysis.

To analyze the effect of *PRKD2*-knockdown on insulin signaling in peripheral tissue, the mouse liver cell line, Hepa1-6 (China Infrastructure of Cell Line Resources, Beijing, China), was cultured at 37 °C under 5% CO₂ and 95% O₂ in Dulbecco's modified Eagle's medium containing 10% heat-inactivated FBS, 100 U/ml penicillin, 100 mg/ml streptomycin (Gibco, Grand Island, NY, USA), and 25 mM glucose. *PRKD2* siRNA was used to knock down *PRKD2* in the same way as

for Ins-1 cells. Insulin signaling activation was determined by western blot of the Akt phosphorylation in proteins collected from Hepa1-6 cells treated with insulin for 15 min.

**Western blotting.** Total protein from mouse tissue samples and cultured cells were extracted using RIPA lysis buffer (50 mM Tris HCl, pH 8.0, 150 mM NaCl, 1% NP-40, 0.5% sodium deoxycholate, and 0.1% sodium dodecyl sulfate (SDS)) supplemented with a protease inhibitor cocktail (Sigma), 10× phosphorylation protease inhibitor (Roche), and 2 mM phenylmethane sulfonyl fluoride. After homogenization with a polytron, the homogenate was incubated on ice for 30 min, and centrifuged at 12,000 g for 15 min at 4 °C. Protein samples (50–100 μg) were separated on 10% SDS-polyacrylamide gel electrophoresis. Blots were incubated at 4 °C overnight with primary antibodies, Rabbbit anti-Akt (1:1000, CST, #4691); Rabbit anti-Phospho-Akt Ser473 (1:1000, CST, #4058L); Mouse anti-PKD2 (1:500, Abcam, ab7281); Mouse anti-PKD2 (1:500, Abcam, ab57114); Rabbit anti-β-actin (1:1000, CST, #8457); Mouse anti-α-Tubulin (1:1000, Sigma, T5168), followed by anti-rabbit or mouse horseradish peroxidase-labeled secondary antibodies (1:1000; Santa Cruz Biotech, sc-2004, sc-2005) for 1 h at room temperature. An electro-chemiluminescence detection system revealed the peroxidase label and relative abundance was quantified by densitometry using Quantity One ver. 4.6.7 software (both from Bio-Rad). Uncropped western blots in this study are shown in Supplementary Fig. 9.

**Histology and immunohistochemistry of pancreas.** Each PFA-fixed pancreas was embedded in paraffin, and 5-μm sections were cut and stained with hematoxylin-eosin (HE) for histological analysis. The HE-stained sections were imaged with a BX51 light microscope and Image-Pro MC 6.0 software (Olympus, Japan) to measure the areas of at least 15 randomly selected islets from each mouse. For Immunohistochemistry, sections were incubated with rabbit anti-insulin (1:400; Santa, sc9168) or mouse anti-glucagon (1:400; Santa, sc57171) antibodies overnight at 4 °C, followed by Alexa Fluor®594 donkey anti-rabbit (1:200; Invitrogen, A21207) or Alexa Fluor®488 goat anti-mouse secondary antibody (1:200; Invitrogen, A11029) for 1 h at room temperature. The slides were imaged with a Zeiss LSM700 confocal microscope. Cells positive for insulin was counted using Image-Pro MC 6.0 software.

**Real-time PCR.** Total RNA was extracted from *PRKD2*$^{-/-}$ or WT β-cells (TIANGEN Biotech, Beijing). Reverse transcription was performed by the M-MLV Reverse Transcriptase (Promega). The primers used for *CACNA1c* amplification were reported in previous study (forward 5′-GGCATCACCAACTTCGACA and reverse 5′-TACACCCAGGGCAACTCATA)[58]. Primers used for β-actin amplification: forward 5′-CACCATGAAGATCAAGATCATTGCT and reverse 5′-AACGCAGCTCAGTAACAGTCCG.

**Nifedipine treatment in rhesus monkeys.** To investigate the inhibition function of nifedipine in insulin secretion in monkeys, we anesthetized the monkeys with ketamine at 10 mg/kg body weight after 14–16 h of fasting. Nifedipine solution was prepared by adding 2 mg nifediping to 5 ml ethylalcohol, 25 ml polyethylene glycol, and 20 ml saline solution, for a final concentration of 40 μg/ml[59]. After baseline blood sampling and blood pressure measurements, Nifedipine was infused at a rate of 5 μg/kg/min for 11 min. Then the infusion rate was changed to 0.5 μg/kg/min for the rest time of the experiment. The IVGTT was started 3 min after changing the infusion rate.

**Statistical analyses.** Statistical analyses in this study were performed using GraphPad PRISM ver. 5.01 (GraphPad Software, Inc.) and the SPSS 18.0 software package (SPSS Inc.). For the animal studies, sample size was determined based on previous study[60]. For in vivo studies in mice, *PRKD2*$^{-/-}$ mice and WT controls were used as different experimental groups. The investigator was blinded to the group allocation during the experiment. Data sets were tested for normality with Kolmogorov–Smirnov tests. Data groups (two groups) with normal distributions were compared using two-sided, unpaired Student's *t*-tests. Data groups with multi-factors were compared using two-way ANOVA tests. A *p*-value <0.05 was considered as statistically significant.

**Data availability.** The raw deep sequencing data in this study are available at SRA under accession numbers SRP070922 and SRP071778. The rest of the data is available from the authors upon reasonable request.

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

## Acknowledgements

We thank Prof. Heping Cheng (Peking University), Prof. Liping Wei (Peking University), Prof. IC Bruce (Peking University) and Prof. Bertrand Chin-Ming Tan (Chang Gung University) for constructive comments on the manuscript. This work was supported by the National Science and Technology Major Projects for "Major New Drug Innovation and Development" (2013ZX09501014), the National Key Basic Research Program of China (2013CB531200), the National Natural Science Foundation of China (81471063, 31221002, 81270883, 31522032, 30870996, and 81370234), the Shenzhen Basic Research Foundation (JCYJ20160428154108239) and the Guangdong Province Basic Research Foundation (2016A020216003).

## Author Contribution

X.Z., C-Y.L., and R-P.X. conceived the idea and designed the study. Y.X. and C.W. performed most of the experiments, and analyzed and interpreted data. J-Y.C., F.L., J.W., N.H., X.H., F.Z., D.M., X.S., Y.D., Y.Z., W.Z., Y.L., H.S., W.Z., C.H., Y.Z., K.O., L.C., and J.C performed part of the experiments and data collection. Y.X. and C.W. wrote the original draft. X.Z., C-Y.L, J-Y.C., and R-P.X. helped with editing. All authors discussed the results and commented on the manuscript.

## Additional information

**Competing interests:** The authors declare no competing interests.

