## [Peer Review File · Nature Communications]

Reviewers' comments:

Reviewer #1 (Remarks to the Author):

This paper reports a somewhat interesting finding of mild obesity and hyperinsulinemia in Prkd2 null mice. The KO of this specific gene was motivated by findings in rhesus monkeys that the paper interprets as showing that a heterozygous truncating mutation of the rhesus orthologue causes primary hyperinsulinism that leads to marked insulin resistance. I would not rule out the possibility that this interpretation may end up being correct but the evidence presented for a causal role in the monkey phenotype falls far short of the most elementary standards of genetic analysis.

Validity concerns

1. First, the reasoning that led to the identification of the PKD2 mutation assumes that the cause of the monkey phenotype is monogenic. I would not argue against a genetic cause but what justifies the assumption that polygenic inheritance has been ruled out? In the vast majority of humans, the metabolic syndrome runs in families as a complex trait, including some cases with extreme phenotype. Simple concordance of the phenotype in two parent-offspring pairs is quite common with complex but highly heritable genetic traits, such as the metabolic syndrome.
2. Even this trivial co-segregation of the mutation with the phenotype is quite tenuous and questionable. Fasting insulin and HOMA-IR in the two mutant offspring was an order of magnitude lower than that of their transmitting fathers at a comparable age (Fig. 1e vs. 3h and 3j, also Fig. 1c vs. 4a and 4c) and indistinguishable from the normal controls (Fig. 1). Hyperinsulinemia is seen only post-glucose. This much milder and qualitatively different phenotype in the offspring actually argues strongly for inheritance as a complex trait.
3. Even assuming monogenic inheritance and assuming that the much milder offspring phenotype represent variable expressivity, the variant filtering that led to a single gene, PKD2, is seriously flawed. It is based on the assumption that the mutation must be truncating, justifying the exclusion of missense mutations. The logic is not obvious, as the majority of autosomal dominant disease is actually due to missense mutations. The exome of a human contains an average of about two hundred low-frequency missense mutations. In the wild, Rhesus monkeys are three time more diverse than humans (Yuan et al., BMC Genet. 2012 Jun 29;13:52. PMID: 22747632). We are not given any information about the genetic diversity of the specific colony but it is clear that a large number (hundreds) of alternatives to PKD2 were overlooked.
4. True, PKD2 shows linkage to the disease but so does one quarter of the genome, after only two meioses. The sample size is totally inadequate for linkage to have any meaning in either proving monogenic inheritance or pinpointing the gene mutated.
5. Ultimately, the only convincing finding in favour of PKD2 is the mild obesity and corresponding hyperinsulinemia in the null mouse. What would have made it interesting, would have been evidence that the hyperinsulinemia is primary and the obesity and insulin resistance secondary rather than the reverse (contrary to what is claimed in the Introduction, a large body of literature has established that the causal relationship insulin resistance - hyperinsulinemia can go either way). It is arbitrarily assumed that the hyperinsulinemia is primary and the obesity and insulin resistance secondary to it. On what basis? Mouse PKD2 is expressed mostly in marrow-derived cells, with extremely low levels of expression in the pancreas (<http://biogps.org/#goto=genereport&id=101540>). The only evidence in favor of a primary effect on the beta cells is the in vitro work with siRNA KD in Ins-1 cells.

Minor points

6. The correct name of the gene is PRKD2. PKD2 is the gene mutated in dominant polycystic kidney disease.

7. The paragraph "The serine/threonine protein kinase D (PKD) . . .insulin secretion." does not belong in the Introduction and should be moved to the Discussion. It would have been appropriate in the introduction only if PKD2 had been examined as a functional candidate gene, which is definitely not the case.

8. Although the English is quite acceptable, someone familiar with proper genetic terminology should go over the paper, to correct awkward expressions. Examples: ". Exome DNA derived from blood cells" and "hereditary".

Reviewer #2 (Remarks to the Author):

In this report Xiao et al. detail 2 Rhesus monkeys that exhibit hyperinsulinemia and identify mutations in the PKD2 gene. The authors go on to show clinical parameters of offspring of one of these animals and the phenotype of mice that lack the PKD2 gene. Mechanistically, Xiao et al. show that loss of PKD2 results in hyper-secretion of insulin in islets ex vivo.

The main premise on which the narrative arc of the paper rests is that hyperinsulinemia can drive insulin-resistance and that therapeutic targets rest in understanding this. I find the underlying data of the paper very confusing in this context. In the monkeys, one is lead to believe that hyperinsulinemia is causing a MetS phenotype, so loss of PKD2 would be expected to be a bad thing. However, throughout the paper it seems to be implied that loss of PDK2 could be thought of as a therapeutic target because it's loss increased glucose stimulated insulin secretion. The primates with loss of PKD2 function have a metabolic syndrome phenotype however. The rodent data appears more consistent with this therapeutic hypothesis, but disproves the hyperinsulinemia-causes-IR hypothesis. This makes for an incoherent and inconsistent picture and would need to be kept consistent throughout the paper or need to be addressed why it isn't consistent.

Perhaps more troubling to me is the evaluation of the individual monkeys. The analysis of 2 primates seems statistically reckless. Perhaps as a discovery dataset to id the putative gene, but the large number of parameters shown with assumption that the data can be interpreted to be the result of the single mutation does not feel appropriate. A thorough explanation and statistical justification for this form of analysis would be required, which would include a clear representation of the distribution of the MetS group of primates.

The evaluation of the offspring animals is also very suspect – showing the distribution of larger numbers of animals on these parameters is important. When you show HOMA-IR I would also like to see the insulin and glucose values.

The rodent data show an animal with improved glucose tolerance but the authors propose that the animals are insulin resistant based on intracellular signaling. It can't be both, one would expect the in vivo data (physiological evaluation of insulin action) to be honored as accurate, bringing into question the interpretation of insulin stimulated insulin resistance.

Reviewer #3 (Remarks to the Author):

In a tour-de-force study, Zhang and colleagues have used an 'extreme phenotype' strategy and deep sequencing to identify a PKD2 mutation in rhesus monkeys that leads to hyperinsulinism that in turn causes insulin resistance. They show that the residue that is mutated (K410X) is conserved and that ablation of PDK2 in mice also results in hyperinsulinism.

I was rather impressed by this paper. Overall, it is clearly and logically written and the data appear convincing. It certainly points in the direction of insulin resistance being caused by hyperinsulinism (rather than the reverse) and this is an important advance that will hugely impact on the field.

There are few things that would make this nice study even stronger:

1) The authors remark that K410 is conserved 'across species ranging from zebrafish to humans'. Are the human insulin resistant patients with the same mutations. unless I miss something, it should not be too difficult to address this question?!

2) Whereas the genetics is top-notch, the functional characterization is perhaps not quite as advanced. Just showing that the amplitude of glucose- and high $[K^+]_o$ -induced $[Ca^{2+}]_i$ transients is increased is not entirely satisfactory. It would be straightforward to determine which calcium channel gene is increased. Given the effects in the heart, it seems possible that it would be the α_1C subunit that is increased (and the same channel plays a critical role in insulin secretion in both mouse and human islets). This would be quite easy to address by quantitative PCR analysis. Stronger still would be patch-clamp analyses of voltage-gated Ca^{2+} currents with pharmacological isolation of the different Ca^{2+} current components. I don't agree that the mechanisms should be the subject of 'future investigation'

3) If it is α_1C that is increased, then it should be possible to counteract the hyperinsulinism (and insulin resistance) pharmacologically using calcium channel blockers (many of which are in clinical use). If they could demonstrate this in the monkeys, it would make the study 'perfect'.

Minor

Fig. 6e. The time of glucose addition should be indicated.

Fig. 6i. Time scale should be given in seconds - is 1800 equivalent to 900 s (difficult to tell from the way data is presented as '500ms/frame')

Reviewers' comments

Reviewer #1 (Remarks to the Author):

This paper reports a somewhat interesting finding of mild obesity and hyperinsulinemia in Prkd2 null mice. The KO of this specific gene was motivated by findings in rhesus monkeys that the paper interprets as showing that a heterozygous truncating mutation of the rhesus orthologue causes primary hyperinsulinism that leads to marked insulin resistance. I would not rule out the possibility that this interpretation may end up being correct but the evidence presented for a causal role in the monkey phenotype falls far short of the most elementary standards of genetic analysis.

We appreciate Reviewer #1's constructive comments. Briefly, we assumed the monogenic inheritance model and hypothesized that some rare variations with large effects (such as nonsense variation, frame-shift or splice-site variation) may contribute to the extreme metabolic phenotype. As a proof-of-concept, the monkey study reveals a heterozygous truncating mutation on *PRKD2* may be a candidate mutation underpinning the primary hyperinsulinism in monkey, which further prompted us to study the metabolic phenotypes of mice *with PRKD2-KO*, a previously neglected candidate gene in metabolic diseases.

Following this reviewer's and the editor's insightful suggestions, we have toned down our claims of heritability, removed the offspring data due to limited linkage evidence in the genetic part, and deleted the claims of causality. Subsequently, we used the monkey data merely for the identification of candidate genes in the revised manuscript (**Page 8, paragraph 1 and 2; Page 14, paragraph 2**).

Validity concerns

1. First, the reasoning that led to the identification of the PKD2 mutation assumes that the cause of the monkey phenotype is monogenic. I would not argue against a genetic cause but what justifies the assumption that polygenic inheritance has been ruled out? In the vast majority of humans, the metabolic syndrome runs in families as a complex trait, including some cases with extreme phenotype. Simple concordance of the phenotype in two parent-offspring pairs is quite common with complex but highly heritable genetic traits, such as the metabolic syndrome.

Although the metabolic syndrome was considered as a “common disease” stemming from the additive effects of common variants, some rare variations with large effects were also proven to account for the phenotypes. As a proof-of-concept study with only two macaque animals showing extreme hyperinsulinemia, we tested whether such rare

variations with large effects do exist underpinning the extreme phenotype. Such a simplified monogenic model led to the identification of one nonsense mutation on *PRKD2* genes as a candidate.

However, we do agree with the reviewer that the polygenic inheritance model could not be ruled out in such an occasion. In the revised version of the manuscript, we toned down the claims of heritability in the genetic part and used monkey data only for the identification of candidate genes (**Page 8, paragraph 1-2**). We also included more discussion on this important issue in the revised manuscript (**Page 14, paragraph 2**).

2. Even this trivial co-segregation of the mutation with the phenotype is quite tenuous and questionable. Fasting insulin and HOMA-IR in the two mutant offspring was an order of magnitude lower than that of their transmitting fathers at a comparable age (Fig. 1e vs. 3h and 3j, also Fig. 1c vs. 4a and 4c) and indistinguishable from the normal controls (Fig. 1). Hyperinsulinemia is seen only post-glucose. This much milder and qualitatively different phenotype in the offspring actually argues strongly for inheritance as a complex trait.

Your points are well taken. But we wish to clarify that the phenotype of extreme fasting hyperinsulinemia seems to be late-onset according to the long-term clinical follow-up study on these macaque animals (**Figure 1a**). The data of fasting insulin and HOMA-IR in the two mutant offspring (**Fig. 1e vs. 3h and 3j, also Fig. 1c vs. 4a and 4c**) were measured at significantly **younger ages** (at 2 years and 4 years) than their transmitting father (which underwent follow-up at age of 12 and the ensuing for 40 months). In the revised manuscript, following the comments of the reviewer as well as the editor's suggestions, we have now removed the offspring data in the genetic part.

*3. Even assuming monogenic inheritance and assuming that the much milder offspring phenotype represent variable expressivity, the variant filtering that led to a single gene, *PKD2*, is seriously flawed. It is based on the assumption that the mutation must be truncating, justifying the exclusion of missense mutations. The logic is not obvious, as the majority of autosomal dominant disease is actually due to missense mutations. The exome of a human contains an average of about two hundred low-frequency missense mutations. In the wild, Rhesus monkeys are three time more diverse than humans (Yuan et al., *BMC Genet.* 2012 Jun 29;13:52. PMID: 22747632). We are not given any information about the genetic diversity of the specific colony but it is clear that a large number (hundreds) of alternatives to *PKD2* were overlooked.*

We appreciate the reviewer for pointing out this important issue and agree that missense variations could also account for the heritability underpinning complex diseases. When considering those missense variations shared by the two macaque animals with extreme phenotypes but not by the other monkeys, an additional 28 missense variations on 24 genes were identified (**Extended Table 2**). Since the nonsense variations have typically stronger effects on the proteome by directly obstructing the open reading frames of proteins, we focused on the nonsense variation on *PRKD2* gene for subsequent experimental verifications.

In the revised manuscript, we have improved the pipelines by including the missense variations and justified the selection of the *PRKD2* nonsense mutation for subsequent experimental verification (**Page 8, paragraph 1-2; Page 14, paragraph 2**).

4. True, PKD2 shows linkage to the disease but so does one quarter of the genome, after only two meioses. The sample size is totally inadequate for linkage to have any meaning in either proving monogenic inheritance or pinpointing the gene mutated.

We agree with the reviewer that the small sample size of offspring monkeys is inadequate for a linkage study to efficiently pinpoint the causal mutations. Knock-in monkeys with this mutation would be the most straightforward means to verify the causality between the *PRKD2* nonsense mutation and the extreme metabolic phenotype, but remain an expensive, time consuming and technically challenging option in monkeys. However, the monkey data prompted us to study the metabolic phenotypes of mice with *PRKD2-KO*.

Thanks to Reviewers' and the editor's suggestions, we removed the offspring data in the genetic part and deleted the claims of causality. Instead, we use monkey data merely for the identification of candidate genes in the revised manuscript (**Page 8, paragraph 1 and 2**).

5. Ultimately, the only convincing finding in favour of PKD2 is the mild obesity and corresponding hyperinsulinemia in the null mouse. What would have made it interesting, would have been evidence that the hyperinsulinemia is primary and the obesity and insulin resistance secondary rather than the reverse (contrary to what is claimed in the Introduction, a large body of literature has established that the causal relationship insulin resistance - hyperinsulinemia can go either way). It is arbitrarily assumed that the hyperinsulinemia is primary and the obesity and insulin resistance secondary to it. On what basis? Mouse PKD2 is expressed mostly in marrow-derived cells, with extremely low levels of expression in the pancreas (<http://biogps.org/#goto=genereport&id=101540>). The

only evidence in favor of a primary effect on the beta cells is the in vitro work with siRNA KD in Ins-1 cells.

We highly appreciate your insightful comments. To address your concerns, we further quantified the insulin levels and activities of the insulin signaling in 4-week old mice, and found that the insulin level was increased in *PRKD2*^{-/-} mice as compared to WT mice, but the body weight and insulin-stimulated Akt phosphorylation were comparable in the two groups (**Extended Fig. 4, Page 9, paragraph 2; Page 21, paragraph 2**), in addition to our *in vitro* work with siRNA KD in Ins-1 cells and Hepa1-6 cells (**Extended Fig. 6**). At 14 weeks of age, both the insulin level and the body weight became significantly higher, while insulin-stimulated Akt phosphorylation was significantly decreased in *PRKD2*^{-/-} mice (**Extended Fig. 5; Fig. 4**). These *in vivo* as well as *in vitro* data indicate that the development of obesity and insulin resistance following hyperinsulinemia. In the revised manuscript, we have included the *in vivo* data to solidify our conclusion that the hyperinsulinemia is primary and the obesity and insulin resistance secondary rather than the reverse.

Minor points

6. The correct name of the gene is PRKD2. PKD2 is the gene mutated in dominant polycystic kidney disease.

Thank you for pointing out this issue. Now we have used the official symbol of this gene (*PRKD2*) in the revised manuscript.

7. The paragraph “The serine/threonine protein kinase D (PKD) . . . insulin secretion.” does not belong in the Introduction and should be moved to the Discussion. It would have been appropriate in the introduction only if PKD2 had been examined as a functional candidate gene, which is definitely not the case.

We revised the manuscript accordingly (**Page 13, paragraph 2**).

8. Although the English is quite acceptable, someone familiar with proper genetic terminology should go over the paper, to correct awkward expressions. Examples: “. Exome DNA derived from blood cells” and “hereditary”.

Thanks to your suggestions, we proofread the manuscript and corrected the typos and non-standard use in the revised version.

Reviewer #2 (Remarks to the Author):

In this report Xiao et al. detail 2 Rhesus monkeys that exhibit hyperinsulinemia and identify mutations in the PKD2 gene. The authors go on to show clinical parameters of offspring of one of these animals and the phenotype of mice that lack the PKD2 gene. Mechanistically, Xiao et al. show that loss of PKD2 results in hyper-secretion of insulin in islets ex vivo.

The main premise on which the narrative arc of the paper rests is that hyperinsulinemia can drive insulin-resistance and that therapeutic targets rest in understanding this. I find the underlying data of the paper very confusing in this context. In the monkeys, one is lead to believe that hyperinsulinemia is causing a MetS phenotype, so loss of PKD2 would be expected to be a bad thing. However, throughout the paper it seems to be implied that loss of PDK2 could be thought of as a therapeutic target because its loss increased glucose stimulated insulin secretion. The primates with loss of PKD2 function have a metabolic syndrome phenotype however. The rodent data appears more consistent with this therapeutic hypothesis, but disproves the hyperinsulinemia-causes-IR hypothesis. This makes for an incoherent and inconsistent picture and would need to be kept consistent throughout the paper or need to be addressed why it isn't consistent.

We agree with the reviewer that a coherent hypothesis should be made for the functional implications of *PRKD2* in metabolic disorders. Overall, our data have provided multiple lines of evidence that the hyperinsulinemia is primary and the obesity and insulin resistance secondary rather than the reverse in both monkey and mouse models.

Specifically, we have shown that:

1. As for the *PRKD2* mutant monkeys, the phenotype of hyperinsulinemia and IR have appeared for more than ten years in these animals, and now one of these monkeys had already developed to severe type-2 diabetes and showed little or no insulin secretion after glucose challenge (**See attached figure for Reviewer**).

2. In the *PRKD2*^{-/-} mice as young as 4 weeks of age, blood insulin level was significantly higher relative to WT mice, while the body weight and insulin-stimulated Akt phosphorylation were comparable at this early time point (**Extended Fig. 4; Page 9, paragraph 2; Page 21, paragraph 2**). Notably, at 14 weeks of age, the body weight and insulin stimulated Akt phosphorylation were altered in *PRKD2*^{-/-} mice as compared to WT mice (**Extended Fig. 5; Fig. 4**). These data indicate that *PRKD2* deficiency primarily increases insulin secretion, and secondarily leads to obesity and insulin resistance.

Although our data supports the hyperinsulinemia-causes-IR hypothesis, short-term *PRKD2* inhibition or deficiency may have beneficial effect via enhancing glucose-stimulated insulin secretion. Nevertheless, sustained *PRKD2* suppression or deficiency is detrimental. In the revised manuscript, we have discussed this issue (**Page 13, paragraph 2; Page 14, paragraph 1**).

Perhaps more troubling to me is the evaluation of the individual monkeys. The analysis of 2 primates seems statistically reckless. Perhaps as a discovery dataset to id the putative gene, but the large number of parameters shown with assumption that the data can be interpreted to be the result of the single mutation does not feel appropriate. A thorough explanation and statistical justification for this form of analysis would be required, which would include a clear representation of the distribution of the MetS group of primates.

Based on the Reviewers' and the editor's constructive suggestions, we have deleted the claims of causality and heritability in the genetic part of the manuscript. In the revised version, we used monkey data merely for the identification of candidate genes (**Page 8, paragraph 1- 2; Page 14, paragraph 2**).

As MetS is a complex disease with changes of multiple metabolic parameters, we have included these metabolic parameters as a supplement to the primary phenotype of hyperinsulinism (**Page 5, paragraph 2**).

The evaluation of the offspring animals is also very suspect – showing the distribution of larger numbers of animals on these parameters is important. When you show HOMA-IR I would also like to see the insulin and glucose values.

We agree with the reviewer that the metabolic parameters of the offspring animals with extreme phenotypes should be illustrated in the context of a larger numbers of macaque animals. However, in the revised manuscript, following Referees' and the editor's suggestions, we removed the offspring data due to its limited statistical power for a linkage study.

The rodent data show an animal with improved glucose tolerance but the authors propose that the animals are insulin resistant based on intracellular signaling. It can't be both, one would expect the in vivo data (physiological evaluation of insulin action) to be honored as accurate, bringing into question the interpretation of insulin stimulated insulin resistance.

To address your concerns, we have performed *in vivo* insulin tolerance test (ITT) and verified the existence of insulin resistance in *PRKD2*^{-/-} mice. Blood glucose level was higher in *PRKD2*^{-/-} mice than that in WT controls after insulin injection (**Fig. 4i and j; Page 9, paragraph 2; Page 21, paragraph 2**), supporting the existence of insulin resistance in *PRKD2*^{-/-} mice. The slightly improved glucose tolerance in *PRKD2*^{-/-} mice is likely due to hyper-secretion of insulin in *PRKD2*^{-/-} mice. We included more descriptions to clarify this issue in the revised manuscript (**Page 9, paragraph 3**).

Reviewer #3 (Remarks to the Author):

In a tour-de-force study, Zhang and colleagues have used an 'extreme phenotype' strategy and deep sequencing to identify a PKD2 mutation in rhesus monkeys that leads to hyperinsulinism that in turn causes insulin resistance. They show that the residue that is mutated (K410X) is conserved and that ablation of PDK2 in mice also results in hyperinsulinism.

I was rather impressed by this paper. Overall, it is clearly and logically written and the data appear convincing. It certainly points in the direction of insulin resistance being caused by hyperinsulinism (rather than the reverse) and this is an important advance that will hugely impact on the field.

We thank the reviewer for your positive assessments on our work. Our responses to your individual concerns are listed below.

There are few things that would make this nice study even stronger:

1) The authors remark that K410 is conserved 'across species ranging from zebrafish to humans'. Are the human insulin resistant patients with the same mutations. unless I miss something, it should not be too difficult to address this question?!

While we found this nonsense mutation in two macaque animals with extreme hyperinsulinism, the human orthologous site is not polymorphic according to the 1000 Genomes Project. However, as we have linked *PRKD2* to obesity and hyperinsulinemia in the null mouse model, it is plausible that patients with loss-of-function mutations on *PRKD2* gene, such as nonsense variation, frame-shift or splice-site variation, should have similar phenotype of hyperinsulinism. In the revised manuscript, we have discussed this important issue (**Page 13, paragraph 1**).

2) Whereas the genetics is top-notch, the functional characterization is perhaps not quite as advanced. Just showing that the amplitude of glucose- and high $[K^+]_o$ -induced $[Ca^{2+}]_i$ transients is increased is not entirely satisfactory. It would be straightforward to determine which calcium channel gene is increased. Given the effects in the heart, it seems possible that it would be the α_1C subunit that is increased (and the same channel plays a critical role in insulin secretion in both mouse and human islets). This would be quite easy to address by quantitative PCR analysis. Stronger still would be patch-clamp analyses of voltage-gated Ca^{2+} currents with pharmacological isolation of the different Ca^{2+} current components. I don't agree that the mechanisms should be the subject of 'future investigation'

We appreciate the reviewer's insightful suggestions and performed additional experiments to address your concerns.

1. We performed real-time PCR analysis to qualify the mRNA level of *CACNA1c* in isolated mouse β cells, and found a significantly increased expression in *PRKD2*^{-/-} mice when compared to WT mice (**Fig. 6e; Page 12, paragraph 1; Page 25, paragraph 3; Page 26, paragraph 1**).

2. Indeed, using patch-clamp analyses, we found that nifedipine-sensitive L-type Ca^{2+} current was significantly greater in β cells from *PRKD2*^{-/-} mice (**Fig. 6c and d; Page 12, paragraph 1**).

These new lines of evidence suggest that enhanced L-type Ca^{2+} channel expression and activity is essentially involved in *PRKD2* deficiency-induced increase of insulin secretion.

3) If it is α_1C that is increased, then it should be possible to counteract the hyperinsulinism (and insulin resistance) pharmacologically using calcium channel blockers (many of which are in clinical use). If they could demonstrate this in the monkeys, it would make the study 'perfect'.

Following your thoughtful suggestions, we treated the *PRKD2* mutant monkey (960109) with nifedipine, since another mutant monkey (950807) now has developed severe type-2 diabetes with little or no insulin secretion in response to glucose challenge (**see attached figure for reviewer**). Notably, continuous intravenous injection of low dose nifedipine

markedly inhibited basal and the glucose stimulated high insulin secretion during IVGTTs in the mutant monkey (**Fig. 6f; Page 12, paragraph 2**). We did not detect significant changes of the blood glucose level after the nifedipine treatments, possibly due to the saturated insulin level in the *PRKD2* mutant monkey (**Fig. 6g; Page 12, paragraph 2**).

Minor

Fig. 6e. The time of glucose addition should be indicated.

We have added the time point of glucose addition in **Fig. 5e**.

Fig. 6i. Time scale should be given in seconds - is 1800 equivalent to 900 s (difficult to tell from the way data is presented as '500ms/frame')

Now, the time scale was changed to seconds in **Fig. 6a**.

REVIEWERS' COMMENTS:

Reviewer #1 (Remarks to the Author):

The revised version has addressed my main concerns. Placing the monkey data in perspective and toning down the conclusions, makes the paper acceptable scientifically but much less of an exciting breakthrough. It still represents a solid piece of work.

Reviewer #2 (Remarks to the Author):

Xiao et al. have revised their manuscript detailing evaluation of the kinase PRKD2 in primates and mice. This work uses hyperinsulinemia identified in 2 primates to identify a candidate gene truncation event that may be responsible for the hyperinsulinemia, and an evaluation of the phenotype of PRKD2 knockout mice, which exhibit hyperinsulinemia and some modest insulin resistance.

To address concerns of the reviewers the authors have more appropriately removed the discussion of the primate offspring and described the evaluation of the primate phenotypes and genomics as a discovery effort that was confirmed with rodent work. This is an improvement, but also an admission that this was an analysis that can not be evaluated statistically and therefore is not useful as a predictor of future success in similar rare variant analyses.

One point that must be fixed would be to include dot plots of the primate data - this would be very helpful in the readers evaluation of the extreme phenotype of individual animals in the context of the individual reference animals.

The inclusion of the worsened ITT in the PRKD2 ko mice, albeit of very minor magnitude, does support the claims that the mouse phenotype is informative in the context of the hyperinsulinemia of the primates. This is an improvement.

The point the authors emphasize - that hyperinsulinemia is preceding insulin resistance - is valid. However, the presence of insulin-resistance in the absence of hypoglycemia is a peculiar argument. In the primates this is accommodated by the new inclusion of the single animals T2DM phenotype (but again, with the difficulties of significance in 1 animal). In rodents one would expect the challenge with high fat diet to cause progression of hyperinsulinemia and mild insulin resistance to actually result in hyperglycemia or glucose intolerance.

Reviewer #3 (Remarks to the Author):

Revision improved this manuscript. I am pleased to see that the authors have performed patch-clamp analyses of the calcium current amplitude and show that it is increased in Prkd2-deficient beta-cells.

A few aspects might deserve consideration:

page 6, last line: increased [cardiac] contractility. Is Cacna1c expression increased in hearts from Prkd2^{-/-} mice?

page 10: 'Hyperinsulinaemia is typically associated...'. This strong statement needs a reference

(myself I am not so sure!)

Page 11: 'It has been shown that Ca²⁺ regulates....'. Statements should be supported by a reference.

Page 12: '...stimulated insulin secretion in PRK2D mutant monkey...'. Worth pointing out that nifedipine had almost no effect in the 'WT' monkey. Worth an explanatory comment!

Page 23: "To investigate L-type Ca²⁺ currents....'. The authors should state how they identified beta-cells.

Point-by point response to referees

REVIEWERS' COMMENTS:

Reviewer #1 (Remarks to the Author):

The revised version has addressed my main concerns. Placing the monkey data in perspective and toning down the conclusions, makes the paper acceptable scientifically but much less of an exciting breakthrough. It still represents a solid piece of work.

We appreciate the reviewer's positive comments on our revised manuscript.

Reviewer #2 (Remarks to the Author):

Xiao et al. have revised their manuscript detailing evaluation of the kinase PRKD2 in primates and mice. This work uses hyperinsulinemia identified in 2 primates to identify a candidate gene truncation event that may be responsible for the hyperinsulinemia, and an evaluation of the phenotype of PRKD2 knockout mice, which exhibit hyperinsulinemia and some modest insulin resistance.

To address concerns of the reviewers the authors have more appropriately removed the discussion of the primate offspring and described the evaluation of the primate phenotypes and genomics as a discovery effort that was confirmed with rodent work. This is an improvement, but also an admission that this was an analysis that can not be evaluated statistically and therefor is not useful as an predictor of future success in similar rare variant analyses.

We thank you for the kind summary and positive assessments on our revised manuscript. We agree with the reviewer's comment and discussed this important issue in the revised manuscript (**Page 13, paragraph 2**).

One point that must be fixed would be to include dot plots of the primate data - this would be very helpful in the readers evaluation of the extreme phenotype of individual animals in the context of the individual reference animals.

In the revised manuscript, we included the dot plots of the primate data as supplementary figures following the suggestion (**Supplementary Fig. 1 and 2**).

The inclusion of the worsened ITT in the PRKD2 ko mice, albeit of very minor magnitude, does support the claims that the mouse phenotype is informative in the context of the hyperinsulinemia of the primates. This is an improvement.

The point the authors emphasize - that hyperinsulinemia is preceding insulin

resistance - is valid. However, the presence of insulin-resistance in the absence of hypoglycemia is a peculiar argument. In the primates this is accommodated by the new inclusion of the single animals T2DM phenotype (but again, with the difficulties of significance in 1 animal). In rodents one would expect the challenge with high fat diet to cause progression of hyperinsulinemia and mild insulin resistance to actually result in hyperglycemia or glucose intolerance.

We agree with the reviewer that further studies with high fat diet challenge may provide additional information to interpret the phenotypes detected in *PRKD2*^{-/-} mice. While as times are needed to obtain enough number of mice and to treat them with high fat diet, we propose to perform this experiment in future studies. In the revised manuscript, we included some discussions on this constructive suggestion (**Page 9, paragraph 2**).

Reviewer #3 (Remarks to the Author):

Revision improved this manuscript. I am pleased to see that the authors have performed patch-clamp analyses of the calcium current amplitude and show that it is increased in Prkd2-deficient beta-cells.

We appreciate the reviewer's positive comments on the revised manuscript

A few aspects might deserve consideration:

page 6, last line: increased [cardiac] contractility. Is Cacna1c expression increased in hearts from Prkd2^{-/-} mice?

According to the suggestion, we performed qPCR in heart tissues from WT and *PRKD2*^{-/-} mice, and found that the mRNA levels of *CACNA1c* are increased in *PRKD2*^{-/-} mice (n=7 for each group).

page 10: 'Hyperinsulinaemia is typically associated...'. This strong statement needs a reference (myself I am not so sure!)

Following the suggestions, we updated this sentence and included references

accordingly in the revised manuscript. (**Page 10, paragraph 1**).

Page 11: 'It has been shown that Ca²⁺ regulates....'. Statements should be supported by a reference.

We added a reference in the revised manuscript (**Page 10, paragraph 3**).

Page 12: '...stimulated insulin secretion in PRKD 2 mutant monkey...'. Worth pointing out that nifedipine had almost no effect in the 'WT' monkey. Worth an explanatory comment!

Following the suggestions, in the revised manuscript, we included more descriptions on the effect of nifedipine treatment in control monkeys (**Page 11, paragraph 3**).

Page 23: "To investigate L-type Ca²⁺ currents....". The authors should state how they identified beta-cells.

Following the suggestions, we included more details in Methods to clarify the identification of β -cells (**Page 22, paragraph 2**). Briefly, isolated islet cells were stimulated with a train of depolarizing pulses from a holding potential of -70 mV, then cells exhibited no voltage-dependent Na⁺ currents were β -cells due to the inactivation of Na⁺ channel at the -70 mV holding potential.